# Role of microRNAs in Obesity-Related Kidney Disease

**DOI:** 10.3390/ijms222111416

**Published:** 2021-10-22

**Authors:** Maite Caus, Àuria Eritja, Milica Bozic

**Affiliations:** Vascular and Renal Translational Research Group, Institute for Biomedical Research in Lleida (IRBLleida)-Fundació Dr. Pifarré and RedInRen RETIC, 25196 Lleida, Spain; mcaus@irblleida.cat (M.C.); aeritja@irblleida.cat (À.E.)

**Keywords:** microRNAs, obesity, kidney, hyperlipidemia, lipotoxicity, obesity-related kidney disease, inflammation, therapeutic agents, renoprotection

## Abstract

Obesity is a major global health problem and is associated with a significant risk of renal function decline. Obesity-related nephropathy, as one of the complications of obesity, is characterized by a structural and functional damage of the kidney and represents one of the important contributors to the morbidity and mortality worldwide. Despite increasing data linking hyperlipidemia and lipotoxicity to kidney injury, the apprehension of molecular mechanisms leading to a development of kidney damage is scarce. MicroRNAs (miRNAs) are endogenously produced small noncoding RNA molecules with an important function in post-transcriptional regulation of gene expression. miRNAs have been demonstrated to be important regulators of a vast array of physiological and pathological processes in many organs, kidney being one of them. In this review, we present an overview of miRNAs, focusing on their functional role in the pathogenesis of obesity-associated renal pathologies. We explain novel findings regarding miRNA-mediated signaling in obesity-related nephropathies and highlight advantages and future perspectives of the therapeutic application of miRNAs in renal diseases.

## 1. Introduction

Obesity is a burgeoning global epidemic and represents an important risk factor for the development and progression of chronic kidney disease (CKD). Indeed, the incidence of obesity-associated nephropathies, as one of the complications of obesity, has risen 10-fold over the last years [1]. Increased fat deposition can lead to a systemic and chronic inflammation, alterations in renin–angiotensin–aldosterone system (RAAS), generation of reactive oxygen species (ROS), as well as hemodynamic and morphological changes in the kidney [2,3,4,5]. All these mutually interdependent processes may subsequently lead to a deterioration of kidney function and its progression to end-stage renal disease (ESRD). Regardless of numerous reports associating fat accumulation and lipotoxicity to renal damage, the underlying pathways responsible for the development of obesity-associated renal impairment is not fully understood. 

miRNAs are short non-coding, single stranded RNA molecules who have a critical role in the regulation of gene expression. miRNAs have been reported to be involved in fundamental biological processes, thus playing an essential role in normal organ development and homeostasis [6,7]. Moreover, miRNAs have been identified as important players in a variety of pathophysiological conditions such as cancer, autoimmune diseases, cardiovascular and renal disorders [8,9,10]. Owing to their unique characteristics such as highly conserved nucleotide sequence of small length and known composition, miRNAs represent a potential basis for the development of novel therapies for miRNA-associated diseases.

The present review explores contemporary knowledge on the role of miRNAs in the pathogenesis of obesity-associated nephropathies, as well as signaling messengers involved in miRNAs-mediated responses in the kidney. We summarize the contemporary findings on the use of miRNAs as targets for potential therapeutic intervention, highlighting benefits and future perspectives of the therapeutic employment of these small non-coding RNA molecules in renal diseases. 

## 2. miRNAs: Biogenesis and Mechanism of Action

MicroRNAs (miRNAs) are small endogenous, non-coding, single-stranded RNA molecules with an average length of 22 nucleotides and highly conserved sequences across species [11]. miRNAs have been reported to have a critical role in the regulation of gene expression through a post-transcriptional processing of messenger RNA (mRNA) [12]. However, their involvement in transcriptional gene activation or silencing has also been described [13]. At present, about 2700 mature miRNAs have been identified in humans [14], and the accumulating evidence indicates that the majority of human genes might be regulated by miRNAs [15]. A single miRNA can regulate ~200 mRNA involved in different cellular pathways, and also, one mRNA transcript can be simultaneously regulated by different miRNAs [15,16]. 

### 2.1. Biogenesis of miRNAs

The biogenesis of miRNAs is under strict spatiotemporal control and the dysregulation of any step of their synthesis is associated with many human diseases. The biogenesis process mostly follows the canonical pathway with two RNase enzymes—Drosha and Dicer playing important roles. However, the existence of alternative, non-canonical pathways that are independent of Drosha or Dicer have also been described [13]. The evidence suggests that ~1% of conserved miRNAs is produced through these alternative pathways [17] and some of the non-canonical miRNAs are reported to be involved in various human diseases [13]. In this section of the review, we will focus our attention on the canonical miRNA biogenesis pathway.

miRNA formation starts with the genomic DNA transcription in the nucleus (Figure 1). The process is catalyzed by RNA polymerase II (RNA Pol II) which is responsible for the formation of a primary miRNA transcript (pri-miRNA) with a terminal loop. In the next step, the pri-miRNA is recognized and cleaved by a microprocessor complex, formed by RNase III Drosha and DGCR8 (DiGeorge syndrome critical region 8) cofactor to finally generate a shorter hairpin, called precursor miRNA (pre-miRNA) [18]. Pre-miRNA is exported into the cytoplasm by exportin-5 RanGTP complex to complete its maturation. In the cytoplasm, pre-miRNA becomes recognized by a Dicer-TRBP complex, formed of RNAase III enzyme Dicer and the transactivating response RNA-binding protein, TRBP [17]. This complex cleaves the terminal loop of the pre-miRNA generating a small miRNA duplex of about 22 nt [18,19], with one strand derived from the 5′ (5p) and the other from the 3′ (3p) arm of the hairpin stem. miRNA duplex subsequently loads onto the Argonaute protein 2 (AGO) to form the pre-RISC (RNA-induces silencing) complex [17]. Within the pre-RISC complex, miRNA duplex will unwind and only one strand will function as a “guide” strand leading the mature RISC complex to a nearby complementary sequence, mostly in the 3′-untranslated region (3′UTR) of the target mRNA, which will have a direct effect on the protein translation [17]. The remaining “passenger” strand of the miRNA duplex will produce inactive and less abundant miRNAs, and will subsequently be discarded [17]. It is believed that both arms have the ability to produce mature miRNAs, but which one of the arms will be selected depends on various conditions. This selection is named “arm switching” [20]. 

### 2.2. Mechanisms of miRNA-Mediated Regulation of Gene Expression

miRNAs play important regulatory roles in a variety of cellular processes through a post-transcriptional regulation in the cytoplasm [12]. However, certain mature miRNAs have also been found within the nucleus where they can play roles in transcriptional gene silencing or activation [12]. During the post-transcriptional regulation, base complementarity between miRNA and its target mRNA mainly occurs through the seed region of miRNA and the complementary sequence of the 3′-untranslated region (3′UTR) of mRNA [18]. Seed region is a 6-mer or 7-mer sequence within the first 2–8 or 1–8 bases of the 5′ end of the miRNA, respectively, which can recognize the target mRNAs [21]. Aside from the canonical seed region, an extended seed region has been discovered which represents a 6-mer sequence within miRNA 4–10 positions [21]. This 6-mer sequence partially overlaps with canonical seed and it has been reported that both of them can match to their targets at similarly high rates [21]. Although the most described complementarity has been reported between miRNAs and the 3′UTR region of the mRNA, binding of miRNAs to the 5′UTR or the coding region of the target mRNA has also been observed [10].

Different miRNA mechanisms of action can be distinguished. On the one hand, a perfect match between the miRNA and its target mRNA 3′UTR can lead to a direct mRNA degradation (Figure 1A), while, on the other hand, an imperfect match can cause a translational repression (Figure 1B) and a decrease in protein levels of the target gene [22]. Interestingly, it has been described that in specific conditions miRNAs can also activate translation of target mRNAs (Figure 1C) [23].

Recent data suggest that certain miRNAs could also be found in the nucleus, regulating gene transcription (Figure 1D), either by silencing or activating target genes through miRNA-binding sites within the gene promoters [13]. Although further research is needed to understand the nuclear localization of miRNAs, different mechanisms that could explain this process have recently been proposed: (a) miRNAs are transported back into the nucleus by different proteins or, (b) all processes of certain miRNAs synthesis occur within the nucleus [13,24].

The expression pattern of miRNAs can be cell- and tissue-specific. miRNAs are essential for normal development and organ homeostasis, and they are involved in fundamental cellular processes such as proliferation, differentiation, apoptosis, and metabolism [6,7]. Furthermore, miRNAs are involved in a variety of pathological conditions such as cancer, cardiovascular disorders, autoimmune and kidney diseases [8,9,10].

## 3. miRNAs in Kidney Physiology and Disease

miRNAs have been detected in embryonic and adult kidney tissue and have been shown to play important roles in renal development and homeostasis, while their deregulation has been associated with various kidney diseases. 

miRNA expression profiling studies have found that miR-192, miR-194, miR-204, miR-215, miR-216 [25], miR-146a, and miR-886 [7] were preferentially expressed in the kidney compared with other tissues. Additionally, miRNAs such as let-7a-g, miR-10a/b, miR-21, miR-30a-e, miR-130, miR-143, miR-196a/b, miR-200a, and miR-872 have also been confirmed to be expressed in renal tissue [26]. 

Recent studies suggested an important role of miRNAs in the regulation of renal development. Namely, Dicer-dependent miRNAs such as miR-17 and 106b [27] have been proved to be essential for the nephron survival [28,29], while conditional mutation of the pre-miR processing enzyme Dicer1 in developing kidney epithelium or stroma led to significant defects in nephrogenesis [28]. Several lines of evidence point to the role of various miRNAs in the regulation of kidney homeostasis and structure [6,30]. Bijkerk et al. [31] demonstrated that miR-132 has an important role in the regulation of body water balance [31] and salt-dependent steady-state renin levels [32], therefore participating in the regulation of RAAS. RAAS is essential for the management of renal salt, water conservation, and blood pressure [32]. Importantly, changes in miRNAs expression have been shown to have an influence on the components of the RAAS pathway [33]. miR-466a/b/c/e family has been shown to be regulated by aldosterone (ALDO) and contributes to a negative feedback loop that reduces long-term ALDO signaling. Thus, the miR-466a/b/c/e family protects aldosterone-sensitive tissues from excessive ALDO exposure [33] and plays an important role in kidney homeostasis. Furthermore, miR-6869-5p has also been described as a regulator of the RAAS cascade [34]. In addition, miR-181 and miR-663 have been shown to regulate renin expression in juxtaglomerular cells [35,36,37]. miR-200b and miR-717 are essential for the maintenance of the electrolyte homeostasis [38], while miR-192 regulates sodium transport in renal epithelial cells [39,40].

Renal tubulointerstitial fibrosis (TIF) is a deleterious process characterized by a destruction of kidney parenchyma [41], and represents a key pathway in the progression of CKD [42]. Numerous studies have supported the role of miRNAs in renal fibrosis. For instance, levels of miR-29a family have been shown to decrease in renal tissue affected by fibrosis [8,18,43]. A study from Chau et al. reported that miR-21 was upregulated in renal fibrosis and diabetic nephropathy, while miR-21 knock out mice presented less kidney lesions after unilateral ureteral obstruction (UUO) or ischemia-reperfusion injury [44]. Several other microRNAs such as miR-200 family, miR-214, miR-199, miR-382, miR-133, miR-212 [15], mir-433 and miR-192 [45], miR-184 [8], and miR-30 family [15] have also been demonstrated to be involved in the process of renal fibrosis and/or tubular epithelial–mesenchymal transition (EMT), one of the key mechanisms in the pathogenesis of renal TIF [46,47]. 

Diabetic nephropathy (DN) is a known complication of type 1 and 2 diabetes which could lead to a progressive kidney disease. A variety of miRNAs have been reported to be involved in the pathogenesis of DN such as miR-21, miR-25 and mir-29 families, miR-34a-5p, miR-141, miR-184, miR-370, miR-377, miR-503, let-7 family, miR-93, miR-126, miR-130b, miR-192, miR-424, and miR-146a, etc., as reviewed in [8]. For instance, miR-192 showed an increased expression in glomeruli of diabetic mice [48], while it decreased in renal proximal tubular epithelial cells during fibrosis [49]. In a mouse model of DN, miR-141 decreased in affected kidneys which subsequently led to an increase of TGF-β1 and development of renal fibrosis [50]. miR-503 has been shown to be involved in diabetic endothelial dysfunction [51], and moreover, its overexpression also caused podocyte cell injury [52]. Recent data reported low levels of miR-25 [53] and miR-126 [54] in peripheral blood of diabetic patients. Furthermore, Lv et al. [55] demonstrated a decrease of miR-130b in plasma of DN patients and proposed this miRNA as a potential predictor of DN [55]. Wang et al. [56] described a remarkable decrease of miR-424 in renal tissue affected by type 1 DN compared with healthy kidney, while its upregulation inhibited apoptosis and reduced pathological changes in the kidney [56]. Similarly, mir-146a decreases in glomeruli of diabetic patients, and is involved in the development of glomerular damage and deterioration of renal function [57]. Additionally, miR-377 have been shown to be overexpressed in mouse mesangial cells during DN [58]. 

The role of miRNAs in renal development and physiology, as well as in pathological conditions of the kidney such as renal TIF, acute kidney injury, DN, lupus nephritis, IgA nephropathy, polycystic kidney disease, etc., has already been extensively reviewed elsewhere [7,8,59,60]. Therefore, in this review, we will focus our attention on the potential role of miRNAs in obesity-related kidney disease.

## 4. Obesity-Related Kidney Disease

Obesity is a growing global epidemic and a major health problem in the world. Due to a rapid change in lifestyle, the prevalence of obesity and obesity-associated complications have risen dramatically within the past two decades. It has been demonstrated that obesity is an independent risk factor for the development of CKD and its progression to ESRD [61,62]. Early observation from the Framingham heart study cohort showed an association of high body mass index (BMI) with higher risk of CKD [63]. Obesity can cause morphological and hemodynamic alterations in the kidney [64], that alongside renal inflammation [65] and oxidative stress [4], may lead to deterioration of kidney function and subsequently glomerulosclerosis and renal TIF [64,66]. 

### 4.1. Effects of Adiposity on Kidney

A significant amount of body fat is generally stored in the adipose tissue (AT) in the form of triglycerides [67,68]. As an active metabolic endocrine organ, AT senses changes in systemic energy balance and actively participates in the regulation of energy homeostasis [69], through autocrine and/or paracrine regulatory mechanisms [69]. As soon as the overall AT depot is surpassed, circulating lipids will start accumulating ectopically in non-adipose tissues including kidney, contributing to tissue damage through a process known as lipotoxicity [68,70]. Furthermore, abnormal hypertrophy of AT will lead to alterations in adipocytokine secretion pattern seen as an increase of serum leptin, resistin, and visfatin, as well as decrease of adiponectin [71], which alongside inflammatory and profibrotic parameters could lead to a renal cell dysfunction (Figure 2). A possible mechanism by which abnormal lipid levels may confer to the progression of renal disease was first explained by Moorhead et al., proposing the lipid nephrotoxicity hypothesis [72]. Namely, in the kidney there are certain types of cells particularly sensitive to lipid accumulation such as renal proximal tubular epithelial cells (RPTECs), podocytes, and mesangial cells [73]. Especially RPTECs seem to be most sensitive to lipid overload due to a fact that proximal tubules almost exclusively use fat as their energy source. Indeed, lipids are an important source and reservoir of energy, and have essential roles in intra- and intercellular signaling in virtually all living cells. However, lipid accumulation in cells that do not possess an adequate molecular machinery to handle large lipid cargos [74,75], such as intrinsic renal cells, may cause kidney injury and dysfunction by activating different effector mechanisms such as generation of ROS [2], impairment of renin–angiotensin–aldosterone activity [3], secretion of proinflammatory and profibrotic factors [4], and insulin resistance [3] (Figure 2). These self-perpetuating secondary events subsequently lead to a progressive impairment of kidney structure and function.

### 4.2. Characteristics of Obesity-Related Kidney Disease

Obesity-related kidney disease is characterized by increased kidney weight and hypertrophy of individual nephrons [3,76]. Thus, tubular and glomerular hypertrophy represent two important structural alterations of the nephron associated with adiposity. Particularly, obese subjects show 3-fold increase of glomerular size and an increased formation of new glomerular capillaries [5], possibly due to the glomerular hyperfiltration [5]. An increase of glomerular filtration rate in obesity leads to a higher filtration fraction and renal tubular overload, which may stimulate sodium and water reabsorption in the proximal tubule, and subsequently further increase in the glomerular filtration via tubuloglomerular feedback [5]. Such renal hemodynamic alterations represent the most important pathophysiological basis for the obesity-related kidney injury. 

A large body of evidence demonstrates association between renal lipid accumulation and kidney dysfunction in various animal models of disease, including models of CKD and models of metabolic disease (obesity, diabetes mellitus, and metabolic syndrome) [64,77,78,79,80,81,82,83,84,85,86,87,88,89]. In the mentioned models, damage of RPTECs and glomeruli have been mostly described, due to an excessive deposition of lipids in these cells. In humans, renal lipid accumulation has been characterized in several conditions, such as focal segmental glomerulosclerosis (FSGS) [90], hypertensive nephrosclerosis [91], minimal change disease (MCD) [92], Fabry’s disease [93], and lipoprotein glomerulopathy [94]. Furthermore, obesity and renal lipid accumulation are essential for the onset and progression of DN in type 2 diabetes (T2D). Therefore, DN in T2D (T2DN) can also be considered as obesity-associated kidney disease [70].

In spite of increasing reports associating obesity and high lipid loads to renal impairment, the molecular mechanism governing the development of kidney dysfunction is not completely understood. Therefore, it is of unmet need to explore new potential therapeutic pathways to prevent and/or reverse the detrimental effects of obesity and abnormal lipid levels on kidney function. 

## 5. miRNAs and Their Functional Role in Obesity-Related Kidney Disease

Numerous studies have supported the role of miRNAs in diverse renal diseases, yet the knowledge of their role in obesity-associated nephropathy is scarce. Here we describe the contribution of miRNAs to different aspects of obesity-associated renal disease. We depict how miRNAs influence initiation and progression of obesity-related kidney injury. In addition, wherever possible, we describe signaling pathways involved in miRNAs-mediated responses in the kidney (Table 1). 

miR-155 is expressed in diverse cells of the kidney such as tubular cells, endothelial and stromal cells, and its elevated levels have been reported to be associated with different types of CKD. Zheng et al. (2019) demonstrated an important role for miR-155 in obesity-associated nephropathy [95]. Namely, the authors revealed that mice fed a high fat diet (HFD) exhibited a marked increase of renal miR-155 which positively correlated with structural and functional damage of the kidney. Moreover, treatment of renal microvascular endothelial cells with palmitic acid led to an increase of miR-155 expression followed by lipotoxic cell damage, inflammation, and oxidative stress. The authors demonstrate that miR-155 directly targets the 3´-UTR of SHIP1/INPP5D and suppresses its expression in vitro and in vivo inducing renal inflammatory response through the NF-kB pathway [95]. Interestingly, specific inhibition of miR-155 led to a suppression of SHIP1/NF-kB signaling in the kidney and significantly ameliorated diet-induced inflammation, oxidative stress, and renal dysfunction [95]. Similar results were obtained in T2DN, a renal disorder characterized by obesity and renal lipid accumulation. Namely, the authors showed a marked increase of miR-155 expression in human and rat kidneys affected by T2DN [96]. The same group described a similar pattern of expression for miR-146a in T2DN [96]. 

Another piece of evidence on the role of miRNAs in obesity-associated nephropathy is the recent work of Sun et al. (2019) [97]. Namely, the authors demonstrated a marked increase of miR-802 expression in the kidneys of C57BL/6J mice fed a HFD, which positively correlated with renal functional parameters of obese mice, such as serum BUN and creatinine [97]. Consistently, obese patients demonstrated higher circulating levels of miR-802 than lean subjects, which correlated positively with creatinine levels but negatively with creatinine clearance. Interestingly, using ultrasound-based microbubble carrying lentivirus delivery method to silence renal miR-802, the authors confirmed that the inhibition of miR-802 protected against HFD-induced inflammation, macrophage infiltration, fibrosis, and functional kidney damage. Mechanistically, Sun et al. provide the evidence of direct binding of miR-802 to 3′UTR of NF-kB repressing factor (NRF) and confirmed that miR-802/NF-kB/NRF signaling could be one of the molecular mechanisms governing the progression of obesity-related nephropathy. The authors proposed therapeutic benefits of using miR-802 inhibitor and suggest miR-802 as a potential biomarker of renal dysfunction in obese subjects [97]. 

Sun et al. (2016) [98] proposed a protective role for miR-451 in kidney injury associated with obesity. Namely, the authors showed a significant downregulation of miR-451 in the kidneys of obese db/db DN mice and peripheral blood mononuclear cells (PBMCs) of patients with DN. Consistently, mesangial cells treated with high glucose showed a dose dependent downregulation of miR-451 [98]. Sun et al. confirmed that miR-451 directly targets 3′-UTR of LMP7 and suppresses LMP7/NF-kB pathway regulating downstream proinflammatory molecules in mesangial cells. Importantly, overexpression of miR-451 significantly ameliorated glomerular injury, albuminuria and expression of proinflammatory and profibrotic factors in the renal cortex of db/db obese mice. Another piece of evidence on the role of miR-451 in the obesity-induced kidney damage is the study of Fluitt et al. (2020) [99]. The authors used an insulin resistant TallyHo/Jng (TH) mouse model shown to be unsusceptible to the development of renal inflammation, injury and fibrosis and they assessed the role of miR-451 in the initiation of nephropathy induced by HFD feeding. Of interest, prolonged systemic inhibition of miR-451 led to a significant renal hypertrophy, albuminuria, kidney injury, fibrosis, and glycogen deposition, as well as dysregulation of autophagy in TallyHo/Jng (TH) obese mice. Moreover, in vitro experiments confirmed the YWHAZ and CAB39 as direct targets for miR-451 and supported the role for miR-451 in reducing renal tubular damage by enhancing autophagy in obese mice via the mTOR signaling pathway [99]. 

miR-18a-5p was found to be markedly downregulated in kidneys of obese db/db mice with an evident renal glomerular dysfunction and injury [100]. Upregulation of miR-18a-5p by resveratrol treatment in mice led to an increase of autophagy and a decrease of apoptosis in the affected kidney, alleviating kidney damage in obese mice. Consistently, overexpression of miR-18a-5p in podocytes confirmed the in vivo results, proposing this miRNA as a negative regulator of apoptosis via modulation of autophagy. The authors identified the Atactic telangiectasis mutation (ATM) gene as a direct target of miR-18a-5p and proposed that the effect of this miRNA on autophagy and apoptosis might be governed via targeting ATM.

miR-130b has also been investigated in the context of obesity-associated renal disease. Particularly, the authors found a marked increase of miR-130b levels and an accelerated kidney dysfunction in adiponectin KO (adipoKO) mice fed a HFD [101]. AdipoKO mice fed an HFD showed signs of renal hypertrophy, albuminuria, lipid accumulation, and decreased nephrin expression. The authors propose that increased expression of miR-130b in adipoKO mice fed an HFD may contribute to renal lipid accumulation, and subsequently to the progression of renal disease in the absence of renoprotective effect of adiponectin.

miR-21 has been one of the widely investigated pathogenic miRNAs in renal disease, mostly due to its profibrotic characteristics and involvement in the TGF-β1 signaling pathway [110]. Of interest, Morrison et al. (2017) [102] investigated the involvement of miR-21 in obesity-associated renal dysfunction. In vivo, HuCRP transgenic mice fed a lard-based HFD developed obesity accompanied by albuminuria, renal inflammation, injury, and fibrosis. The authors demonstrated that obese mice showed an increased expression of miR-21 in the kidney compared with mice fed a regular chow, which correlated significantly with the expression of renal tubular injury marker Kim-1 and the grade of renal fibrosis. The authors propose PPAR-γ pathway as a possible link in regulating miR-21 levels in obesity-induced nephropathy [102]. 

As stated above, T2DN has also been considered an obesity-associated kidney disease [111], while the renal lipid accumulation and metabolic changes related to obesity have been shown to be essential for the onset and progression of T2DN. One of the animal models frequently used to study the pathological progression of type 2 diabetes and its consequences to other organs, is the high fat diet-streptozotocin (HFD/STZ) rodent model. Thus, Zhao et al. (2021) [103] described the role of miR-365 in nephropathy induced by the HFD/STZ treatment in rats. miR-365 was significantly expressed in diseased renal tissue alongside an elevated serum BUN and creatinine, urinary albumin, and inflammatory markers. The authors proposed that miR-365 had a potential binding site for BDNF and that the increase of miR-365 repressed the protein expression of BDNF and p-TrkB, thereby promoting the kidney damage through BDNF/TrkB signaling axis. Downregulation of miR-365 led to a reduced secretion of inflammatory cytokines and profibrotic markers in high glucose-treated human proximal tubular epithelial (HK2) cells, while increased BDNF expression allowed for alleviation of renal cell damage and dysfunction [103].

Xue et al. (2018) [104] demonstrated an involvement of miR-34a-5p in T2DN induced by HFD/STZ in mice and proposed this miRNA as a promising candidate for the development of a novel therapeutic tool to prevent/treat DN. Namely, miR-34a-5p showed a significant upregulation in the renal tissue of HFD/STZ-induced diabetic mice and high glucose treated HK2 cells, alongside a dramatical increase of profibrotic markers such as collagen, fibronectin, and TGF-β1. The authors provide evidence that miR-34a-5p directly targets 3′UTR of SIRT1 as its genuine target and propose miR-34a-5p/SIRT/TGF-β1 signaling as a crucial in tubulointerstitial damage during T2DN [104]. 

Li et al. (2019) [105] proposed a protective role for miR-26a-5p in kidney dysfunction induced by HFD/STZ treatment. The authors demonstrated that inhibition of miR-26a-5p in HK2 cells promoted inflammatory response in these cells, while its overexpression ameliorated cell dysfunction. Using TargetScan and luciferase reporter assay, the authors confirmed that miR-26a-5p directly targeted the 3′-UTR of CHAC1 in HK2 cells, while subsequent gain- and loss-of-function experiments revealed that miR-26a-5p ameliorated the inflammatory response in renal cells through the CHAC1/NF-kB pathway. Interestingly, the same group showed that miR-26a-5p was significantly decreased in urinary exosomes of T2DN patients [105,106].

Shan and colleagues (2016) [107] assessed the role of miR-10a in extracellular matrix accumulation in the kidney of diabetes mellitus induced by combined treatment of HFD/STZ. They observed that both HFD and HFD/STZ administration decreased levels of miR-10a expression in the mouse kidney. Moreover, tail intravenous injection of miR-10a mimics attenuated the higher urine albumin-to-creatinine (ACR) ratio and reversed the kidney damage induced by HFD/STZ, while silencing of miR-10a elevated the kidney ACR ratio in naive mice. Of interest, Shan et al. demonstrated that miR-10a directly targeted the 3′UTR of CREB1, thus regulating the production and accumulation of extracellular matrix and kidney function in obesity-induced nephropathy. Altogether, the authors propose the HDAC3/miR-10a/CREB1 pathway as a new possible signaling mechanism governing kidney injury in type 2 diabetes.

Matboli et al. (2017) [108] described the role of miR-133b, miR-342, and miR-30a in nephropathy induced by HFD/STZ treatment in rats. Analyzed miRNAs were significantly upregulated in diseased renal tissue alongside an elevated serum lipids and BUN, creatinine clearance, and urinary albumin. Furthermore, the authors identified autophagy genes RB1CC1, MAP1LC3B, ATG-12 as direct targets of miR-133b, miR-342, and miR-30a, respectively. The authors hypothesized that HFD/STZ upregulated miR-133b, miR-342, and miR-30a with subsequent downregulation of autophagy in the kidney leading to renal dysfunction possibly via the AMPK/PI3K pathway [108]. 

miR-214 has been shown to be highly expressed in human renal disease and animal models of kidney disease [112]. Thus, Yan et al. (2019) found miR-214-3p to be upregulated in the kidney and serum of rats treated with a combined treatment of long-term HFD and short-term sodium taurocholic injection [109]. Rats developed renal damage and pancreatitis followed by an inhibition of PTEN expression and an increase of pAkt levels in kidneys. Treatment of rats with anti-miR-214-3p reversed the renal inflammation and fibrosis, as well as expressions of PTEN and pAkt. The authors propose the miR-214-3p/PTEN/Akt pathway as responsible for tissue damage and fibrosis in HFD/sodium taurocholic rat model.

Li et al. [113] showed strong dysregulation of miRNA expression profile in both porcine model and human subjects with obesity and metabolic syndrome (MetS). Namely, delivery of extracellular vesicles produced by adipose tissue mesenchymal stem cells from obese MetS pigs to animals with renovascular disease aggravated senescence and renal fibrosis in injured kidneys [113].

## 6. miRNAs as a New Therapeutic Approach in CKD: Advantages and Perspectives

Given the ample evidence of the involvement of miRNAs in the pathogenesis of various diseases, it is plausible to think that modulation of miRNAs and their function could be used as a therapeutic approach in different renal diseases, including obesity-associated nephropathies. miRNAs play essential roles in the gene regulation and have the ability to inflect numerous gene pathways [10]. Owing to their specific features, such as highly conserved short sequence of known nucleotide composition, miRNAs represent a new attractive class of targets for potential therapeutic mediation [114]. 

Principally, we can distinguish two approaches in the development of miRNA-based therapeutics: (a) inhibition and (b) restoration of miRNA activity/function. The specific miRNA activity can be silenced by using several methods that comprise chemically modified antisense oligonucleotide (ASO) inhibitors or the transgenic introduction of tandem miRNA-binding site repeats (known as Decoy or Sponge technologies) [115,116,117]. Modified antisense oligonucleotides (hereafter called anti-miRs) are composed of full or partially complementary reverse sequence of a mature miRNA and are capable of reducing indigenous levels of specific miRNA. Anti-miR works as a competitive inhibitor of miRNAs and elicits its effects following the annealing to the mature miRNA guide strand after the RNA-induced silencing complex has removed the passenger strand [118]. According to Rooij et al. [119], the essential requirements for a successful and effective anti-miR are (a) cell permeable chemistry; (b) slow excretion; (c) an in vivo stability; (d) high specificity binding to the miRNA of interest [119]. Therefore, several modifications were done in this context so far, such as chemical modifications for stability and cholesterol conjugation for better cellular uptake [116]. Thus, some examples of these approaches are the inhibition of miR-122 by 2´-O-methoxyethyl phosphorothioate antisense oligos [120], cholesterol-tagged 2′-O-Me antisense oligo (antagomir-122) [116], or antisense locked nucleic acid modified oligos (LNA–anti-miR) [121] that ameliorated hypercholesterolemia in mouse models [116,120,121]. Moreover, systemic delivery of LNA–anti-miR-122 led to a long-lasting decrease of total plasma cholesterol in a nonhuman primate model without any evidence for LNA-associated toxicity, thus confirming the potential of modified oligonucleotides as a novel class of therapeutics for disease-associated miRNAs [115]. 

Another described approach for the inhibition of specific miRNA action is the expression of tandem repeats of miRNA-binding sites (Decoy or Sponge) [117,122]. Namely, miRNA sponges contain complementary binding sites to a miRNA of interest, specifically to its seed region. Therefore, according to Ebert et al. miRNA sponges should be able to block a whole family of related miRNAs [117]. Zheng et al. (2019) demonstrated that suppression of renal miR-155 by miR-155 sponge treatment efficiently attenuated HFD-induced renal inflammation, lipotoxicity, macrophage infiltration, as well as structural and functional damage of the kidney induced by obesity in mice [95]. 

In renal pathologies in which miRNAs are downregulated, a potential therapeutic approach would be the reestablishment of miRNA’s function by the administration of miRNA mimic. miRNA mimics are double-stranded RNA molecules which can separate intracellularly to a single-stranded RNA. Subsequently, one strand loads into the RISC and functions as a miRNA [114]. Restoration of expression of several miRNAs such as miR-146, miR-155, miR-451, miR-10a, miR-18a-5p using RNA mimic technology ameliorated significantly renal injury and dysfunction in different in vitro and in vivo models of renal disease [96,98,100,107]. In spite of significant progress of miRNA mimic technology, there are still issues that need to be addressed before using miRNA mimics in clinical practice. Such issues would be an in vivo delivery, dosage, immune response, cellular uptake, and in vivo stability [112].

Targeting miRNAs to the kidney continues to be an important challenge especially if we wish to bypass possible undesirable consequences in other tissues and organs, as well as target-off effects [110]. Despite many examples of successful delivery of mimics and inhibitors to the kidney via intravenous and subcutaneous injections [123], the aspect of targeting miRNAs to the kidney or specific kidney cells, while at the same time avoiding toxicity and adverse effects in other tissues and/or activation of adaptive immune response, still stays an important issue to be considered. Apart from delivery of miRNAs and safety concerns, another aspect that should be taken into consideration while designing miRNA therapies for the kidney disease would be the clearance of these molecules. Currently, there are scarce data from animal studies dealing with this matter. 

Stability of miRNAs is an essential requirement for the miRNA-based therapies. Significant progress has been made to increase RNA stability in vivo by different molecular modifications of the backbone [114]. In this context, development of locked nucleic acid (LNA) technology holds a great promise as LNA-modified oligonucleotides (LNA-anti-miRs) exhibit high binding affinity to complementary RNA target and high stability in vivo and in vitro [115,121]. Furthermore, to augment the affinity for complementary nucleotides, 2′-O-methoxyethyl (2′-MOE) and 2′-oxy-methyl (2′-OMe) modifications have been successfully designed and applied [116,120]. 

## 7. Conclusions

A growing body of evidence indicates that miRNAs play a paramount role in gene regulation and have the capacity to modulate a myriad of gene pathways. It has been demonstrated that miRNAs have essential roles in regulation of normal renal development and physiology, while their aberrant expression has been linked to the development of different renal diseases. Furthermore, miRNAs have emerged as important players in the onset and development of obesity-associated diseases by affecting the status and functions of indigenous renal cells such as RPTECs, podocytes, and mesangial cells during renal lipid accumulation.

RPTECs, podocytes, and mesangial cells seem to be the most sensitive to lipid accumulation due to an absence of the molecular machinery necessary to handle large lipid overloads. Thus, lipid accumulation in these cells will lead to a cell dysfunction and injury by triggering inflammation, oxidative stress, impairment of renin–angiotensin–aldosterone activity, and insulin resistance. It is now clear that high lipid loads and lipotoxicity in the kidney modulate the expression of a variety of miRNAs, and their renal expression significantly correlates with the lipotoxic cell damage, inflammation, oxidative stress, as well as structural and functional damage of the kidney. 

Owing to their unique characteristics, miRNAs have risen as a new attractive class of targets for possible therapeutic intervention in different renal diseases, including obesity-associated renal pathologies. Targeting miRNAs directly to inhibit or reestablish their activity/function could be promising therapeutic strategies. Despite the fact that there are various challenges to be resolved on our journey to a successful therapeutic approach for clinical application, miRNAs still hold a tremendous promise in designing a new generation of therapy for different types of renal diseases.

## Figures and Tables

**Figure 1 ijms-22-11416-f001:**
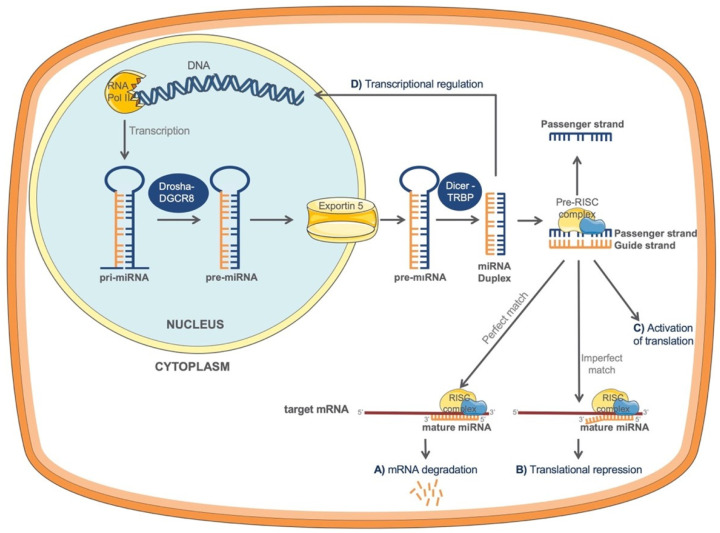
miRNAs biogenesis and post-transcriptional gene regulation mechanisms. In the nucleus, genomic DNA is transcribed by RNA polymerase II to form a pri-miRNA. pri-miRNA is recognized and cleaved by the complex composed of RNase III Drosha and DGCR8 to generate pre-miRNA. Pre-miRNA is exported into the cytoplasm by exportin-5 RanGTP and further cleaved by Dicer-TRBP to generate miRNA duplex. This duplex is loaded onto the pre-RISC complex, where consequently only one strand (“guide strand”) will stay to form the RISC complex. The guide strand will lead the mature RISC complex to a nearby 3′ UTR complementary sequence of the target mRNA. If miRNA and its target mRNA establish the perfect match, a direct mRNA degradation will occur (**A**). Alternatively, an imperfect match between miRNA and its target mRNA will lead to a translational repression and a decrease in protein levels of the target gene (**B**). In certain conditions, miRNAs can activate translation of target mRNA (**C**). miRNAs could also be found in the nucleus regulating gene transcription (**D**), either by silencing or activating target genes through miRNA-binding sites within the gene promoters. DGCR8, DiGeorge syndrome critical region 8 cofactor; TRBP, transactivating response RNA-binding protein.

**Figure 2 ijms-22-11416-f002:**
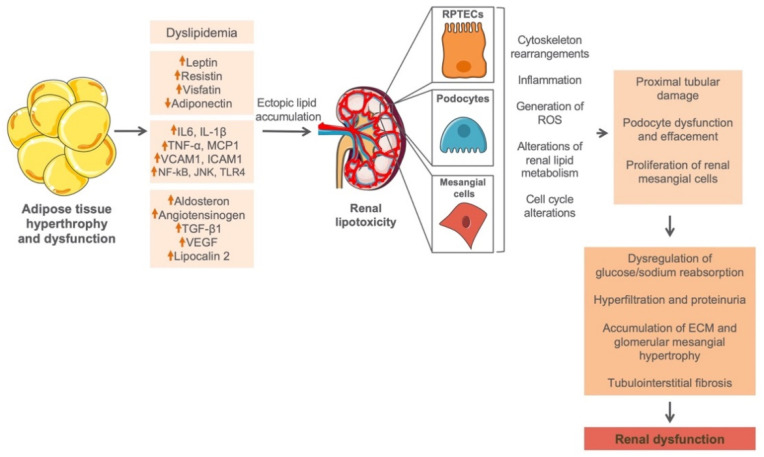
Effects of adiposity on normal renal function. Abnormal hypertrophy of adipose tissue in obesity may lead to changes in expression of different adipocytokines, inflammatory and profibrotic parameters, as well as the ectopic accumulation of circulating lipids in the kidney, contributing to tissue damage through a process known as lipotoxicity. RPTECs, podocytes, and mesangial cells are not equipped with an adequate machinery to handle large lipid overloads; thus, accumulation of lipids in these cells will lead to a cell dysfunction accompanied by the generation of ROS, impairment of renin–angiotensin–aldosterone activity, secretion of proinflammatory and profibrotic factors, and insulin resistance. These self-perpetuating secondary events may subsequently lead to a further damage of renal cells and progressive impairment of kidney structure and function. RPTECs, renal proximal tubular epithelial cells; ROS, reactive oxygen species; ECM, extracellular matrix.

**Table 1 ijms-22-11416-t001:** miRNAs involved in obesity-associated nephropathy.

miRNA	Experimental Model	Expression Pattern	Target Gene	Signaling Pathway	Reference
miR-155	C57BL/6J HFD MECs	Increase	SHIP1/INPP5D	SHIP1/NF-kB	[95]
	T2DN patients HFD/STZ	Increase	n/i	NF-kB	[96]
miR-146a	HFD/STZ	Increase	n/i	NF-kB	[96]
miR-802	C57BL/6J HFD Obese patients	Increase	NRF	NF-kB/NRF	[97]
miR-451	db/db				
	PBMCs GMCs	Decrease	LMP7	LMP7/NF-kB	[98]
	TallyHo/Jng HFD		YWHAZ, CAB39	mTOR	[99]
miR-18a-5p	db/db Podocytes	Decrease	ATM	n/i	[100]
miR-130b	adipoKO HFD	Increase	n/i	n/i	[101]
miR-21	HuCRP HFD	Increase	n/i	PPAR-γ	[102]
miR-365	HFD/STZ HK2	Increase	BDNF	BDNF/TrkB	[103]
miR-34a-5p	HFD/STZ HK2	Increase	SIRT1	SIRT/TGF-b1	[104]
miR-26a-5p	HFD/STZ HK2 T2DN patients	Decrease	CHAC1	CHAC1/NF-kB	[105,106]
miR-10a	HFD/STZ	Decrease	CREB1	HDAC3/CREB1	[107]
miR-133b	HFD/STZ	Increase	RB1CC1	AMPK/PI3K	[108]
miR-342	HFD/STZ	Increase	MAP1LC3B	AMPK/PI3K	[108]
miR-30a	HFD/STZ	Increase	ATG-12	AMPK/PI3K	[108]
miR-214-3p	HFD/STH	Increase	n/i	PTEN/Akt	[109]

HFD, high fat diet; MECs, microvascular endothelial cells; T2DN, diabetic nephropathy in type 2 diabetes; STZ, streptozotocin; PBMCs, peripheral blood mononuclear cells; GMCs, glomerular mesangial cells; HK2, human proximal tubular epithelial cells; STH, sodium taurocholic injection.

## Data Availability

Not applicable.

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
