# Peer review of "Role of microRNAs in Obesity-Related Kidney Disease"

_ijms, 2021, doi:10.3390/ijms222111416_

Round 1

Reviewer 1 Report

The review by Caus M et al., is a very interesting manuscript focused in the role of different miRNAs in the renal alterations associated with obesity. The review show different miRNAs involved in renal pathology and later that is focused in obesity-related kidney disease. Thus, the review highlights a hot topic in the last years and provides important information in the role of miRNAs as potential therapeutic targets for renal diseases. The review is well structured and clearly presented.

For these reasons, I have only minor comments concerning the manuscript:

  1. The authors have mentioned the participation of renin-angiotensin-aldosterone system in the progression of kidney disease. However, the effect of RAAS on miRNAs have not been described. There are few data regarding this, however, I would suggest to include the studies from Ozbaki-Yagan (FASEB J, 2020; PMID: 32652691), Jan van Zonneveld (Commun Biol. 2020; PMID: 32409785), Liu (Front Med, 2021; PMID: 34568382) to clarify this point.
  2. In addition, I would suggest to include the study from Li et al. (Cell Commun Signal; PMID: 32787856) in which the authors show alterations in different miRNAs in patients and in an animal model of obesity and its possible involvement in senescence and renal fibrosis.
  3. Both figures presented by the authors are useful to the readers for the understanding of the review, however, the quality of the images should be improved in terms of higher resolution.
  4. There are some spelling mistakes (for example mir instead of miR in line 182).
  5. Authors should revise the abbreviations used within the manuscript. Several times authors use abbreviations but not always (for example for diabetic nephropathy). Please, homogenize it. In addition, some abbreviations are defined several times.
  6. Line 507, replace ther for their.

Author Response

Reviewer 1

The review by Caus M et al., is a very interesting manuscript focused in the role of different miRNAs in the renal alterations associated with obesity. The review show different miRNAs involved in renal pathology and later that is focused in obesity-related kidney disease. Thus, the review highlights a hot topic in the last years and provides important information in the role of miRNAs as potential therapeutic targets for renal diseases. The review is well structured and clearly presented.

Response: We thank the Reviewer #1 for the positive assessment of our work.

For these reasons, I have only minor comments concerning the manuscript:

Reviewer 1, Comment 1: The authors have mentioned the participation of renin-angiotensin-aldosterone system in the progression of kidney disease. However, the effect of RAAS on miRNAs have not been described. There are few data regarding this, however, I would suggest to include the studies from Ozbaki-Yagan (FASEB J, 2020; PMID: 32652691), Jan van Zonneveld (Commun Biol. 2020; PMID: 32409785), Liu (Front Med, 2021; PMID: 34568382) to clarify this point.

Response: We thank the reviewer #1 for this valuable suggestion. We have now included suggested studies in this review and clarified the relationship between RAAS and miRNAs (Section 3, page 4).

Reviewer 1, Comment 2: In addition, I would suggest to include the study from Li et al. (Cell Commun Signal; PMID: 32787856) in which the authors show alterations in different miRNAs in patients and in an animal model of obesity and its possible involvement in senescence and renal fibrosis.

Response: We have now included the suggested reference and explained results from the study in the Section 5, page 10).

Reviewer 1, Comment 3: Both figures presented by the authors are useful to the readers for the understanding of the review, however, the quality of the images should be improved in terms of higher resolution.

Response: We thank the reviewer for this suggestion. We have now uploaded higher resolution images as separated files.

Reviewer 1, Comment 4: There are some spelling mistakes (for example mir instead of miR in line 182).

Response: We have now corrected mistakes in the Line 182 (new position - Line 251).

Reviewer 1, Comment 5: Authors should revise the abbreviations used within the manuscript. Several times authors use abbreviations but not always (for example for diabetic nephropathy). Please, homogenize it. In addition, some abbreviations are defined several times.

Response: We thank the Reviewer #1 for this valuable suggestion. We have now revised the abbreviations used in the manuscript and made corrections accordingly.

Reviewer 1, Comment 6: Line 507, replace ther for their.

Response: We have now corrected the misspelling in the Line 507 (new position - Line 601).

Reviewer 2 Report

Minor grammatical and syntax corrections:

Line 8 - Obesity is a major (not the major)

Line 28 - fold (not folds)

Line 45 - new fangled (change this)

Line 58 - Rephrase to 'However, their involvement in transcriptional gene activation or silencing has also been described.'

Line 68 - through (not trough)

Otherwise well written, organized and interesting. 

Author Response

Reviewer 2

Minor grammatical and syntax corrections:

We thank the Reviewer #2 for pointing out grammatical and syntax errors. We have now made appropriate corrections as explained in our responses.

Reviewer 2, correction 1: Line 8 - Obesity is a major (not the major)

Response: We have introduced the correction in the Line 8.

Reviewer 2, correction 2: Line 28 - fold (not folds)

Response: We have now corrected the misspelling in the Line 28.

Reviewer 2, correction 3: Line 45 - new fangled (change this)

Response: We have now changed the word suggested by the reviewer.

Reviewer 2, correction 4: Line 58 - Rephrase to 'However, their involvement in transcriptional gene activation or silencing has also been described.'

Response: Thank you for the suggestion. We have now rephrased the Line 58 (new position - Line 60).

Reviewer 2, correction 5: Line 68 - through (not trough)

Response: We have now corrected the misspelling in the Line 68 (new position - Line 72).

Otherwise well written, organized and interesting. 

Response: We thank the Reviewer #2 for the positive appraisal of our work.
